# Prevalence of Undernutrition, Frailty and Sarcopenia in Community-Dwelling People Aged 50 Years and Above: Systematic Review and Meta-Analysis

**DOI:** 10.3390/nu14081537

**Published:** 2022-04-07

**Authors:** Nada Almohaisen, Matthew Gittins, Chris Todd, Jana Sremanakova, Anne Marie Sowerbutts, Amal Aldossari, Asrar Almutairi, Debra Jones, Sorrel Burden

**Affiliations:** 1School of Health Sciences, University of Manchester, Manchester M13 9PL, UK; nada.almohaisen@postgrad.manchester.ac.uk (N.A.); matthew.gittins@manchester.ac.uk (M.G.); chris.todd@manchester.ac.uk (C.T.); jana.sremanakova@postgrad.manchester.ac.uk (J.S.); annemarie.sowerbutts@manchester.ac.uk (A.M.S.); amal.aldossari@postgrad.manchester.ac.uk (A.A.); asrarsalemi.almutairi@postgrad.manchester.ac.uk (A.A.); debra.jones@manchester.ac.uk (D.J.); 2Manchester University Foundation NHS Trust, Manchester M13 9WL, UK; 3Salford Royal Foundation NHS Trust, Salford M6 8HD, UK

**Keywords:** undernutrition, frail, elderly, sarcopenia, prevalence, incidence, nutrients, mortality, morbidity, systematic review, meta-analysis

## Abstract

The world’s population aged ≥65 is expected to rise from one in eleven in 2019 to one in six by 2050. People aged ≥65 are at a risk of undernutrition, frailty, and sarcopenia. The association between these conditions is investigated in a hospital setting. However, there is little understanding about the overlap and adverse health outcomes of these conditions in community-dwelling people. This systematic review aims to quantify the reported prevalence and incidence of undernutrition, frailty, and sarcopenia among older people aged ≥50 living in community dwellings. Searches were conducted using six databases (AMED, CENTRAL, EMBASE, Web of Science, MEDLINE, and CINAHL), and 37 studies were included. Meta-analyses produced weighted combined estimates of prevalence for each condition (Metaprop, Stata V16/MP). The combined undernutrition prevalence was 17% (95% CI 0.01, 0.46, studies *n* = 5; participants = 4214), frailty was 13% (95% CI 0.11, 0.17 studies *n* = 28; participants = 95,036), and sarcopenia was 14% (95% CI 0.09, 0.20, studies *n* = 9; participants = 7656). Four studies reported incidence rates, of which three included data on frailty. Nearly one in five of those aged ≥50 was considered either undernourished, frail, or sarcopenic, with a higher occurrence in women, which may reflect a longer life expectancy generally observed in females. Few studies measured incidence rates. Further work is required to understand population characteristics with these conditions and the overlap between them. PROSPERO registration No. CRD42019153806.

## 1. Introduction

Life expectancy continues to increase worldwide, particularly in people older than 65 years [1]. In 2019, around 9% of the global population was aged 65 years or older, and it is predicted to rise to nearly 12% by 2030 and almost 23% by 2100 [2]. This global increase includes the UK, where people aged 65 years and above are predicted to increase from 18.5% (one in five people) in 2019 to 23.9% (one in four people) in 2039 [3]. It is, therefore, reasonable that promoting healthy ageing is a priority for healthcare providers [4]. Ageing well is an important concept, as the ageing process is related to metabolic, physiological, and functional impairments [1]. In addition, older people have a high prevalence of multiple morbidities that affect overall wellbeing and survival [5]. These include three common conditions, undernutrition, frailty, and sarcopenia, that increase the risk of falling, hospitalisation, loss of independence, and increased morbidity and mortality rates [6].

In the UK, approximately three million people are undernourished [7], and the likelihood of occurrence increases with age and comorbidities [8]. There is growing interest in the link between undernutrition and frailty due to similarities in the criteria used to identify undernutrition and those used to determine frailty [9,10]. Weight loss is used as a criterion for the identification of both undernutrition and frailty. In addition, a decrease in skeletal muscle mass is associated with functional decline, which is a characteristic of both frailty and undernutrition [9,11,12]. A poor nutritional status is strongly associated with frailty and has been demonstrated in many studies among older people in both hospital and community settings [11,12,13].

Frailty is also common in older people and increases with ageing. It is reported to affect 6.5% of older adults aged between 60 and 69 years, rising to 65% in people over 90 years [14]. Similarly, the prevalence of frailty has been reported as 4.1% among people aged 50 to 64 years, and this increased to 17% in those 65 years and over [15]. Prefrailty and frailty may start at middle age, suggesting that a frailty assessment and prevention would be useful early within the fourth decade [15,16]. There is a gender difference in frailty, with more females than males often affected [15,16,17].

Sarcopenia is another age-related condition that can be described as a loss of muscle mass that is associated with impaired strength and reduced activities and physical performance [18]. There is a similarity also between frailty and sarcopenia, which both involve the loss of fat-free mass that leads to lower strength and function in older people [19]. However, sarcopenia and frailty require different therapeutic treatments. People with sarcopenia were more likely to also be frail, albeit those who were frail were less likely to be sarcopenic [20]. The number of people with sarcopenia is predicted to increase to 1.2 billion by 2025 and double by 2050 [21]. Sarcopenia is associated with increased morbidity and mortality in older adults [22].

All three conditions are associated with unfavourable outcomes. Undernutrition can exacerbate undesirable health conditions, increasing the likelihood of morbidity and mortality [23,24]. It can be considered an independent risk factor for impaired mobility and walking instability, leading to an elevated risk of falling [25]. Older people who are undernourished have higher rates of surgical complications, experience longer inpatient admissions [26], have a worsening clinical status [27], and an overall decreased quality of life and life expectancy [28]. Likewise, frailty in older people is associated with an increased risk of morbidity, mortality, falls, dependency, disability, and hospitalisation, as well as higher healthcare costs [14,16,29,30,31,32]. Similarly, sarcopenia leads to impaired strength, lowers daily activity, and reduces physical performance [33]. Older people with sarcopenia have a greater chance of falling, bone fractures, and disability [34]. Consequently, there is an association between sarcopenia and poor quality of life, including physical and mental components, among community-dwelling older people [35,36]. A positive correlation between sarcopenia and cognitive dysfunction in older people has been demonstrated [37], along with poor overall health, resulting in increased healthcare costs [38].

The relationship between undernutrition, frailty, and sarcopenia is vital to consider in the nutritional management of older people living in the community. There are many similarities between the three conditions, and these conditions have wide-reaching implications for functionality, independence, cognitive function, and overall wellbeing [10]. One study investigated the overlap of four related conditions—undernutrition, cachexia, frailty, and sarcopenia—in inpatients aged ≥70, which found that one-third had at least one of these conditions. Moreover, authors concluded that these four conditions overlapped and were interrelated [39]. However, the relationship between the three conditions is still developing, as little is known about their overlap. Hence, this systematic review aims to examine the evidence on the prevalence and incidence of undernutrition, frailty, and sarcopenia in older people aged ≥50 years living independently in community dwellings. In addition, it aims to determine the gender difference in the three conditions. Body composition begins to change in adults over 50 years and above as part of the aging process, which includes a decline in testosterone, oestrogen, and the onset of muscle mass reduction [40]. Therefore, it follows that these three conditions develop earlier in middle age [16,41,42,43].

## 2. Materials and Methods

The protocol was registered on the international prospective register of systematic reviews (PROSPERO) with reference number CRD42019153806 on 24 October 2019. The review followed the Preferred Reporting Items for Systematic Reviews and Meta-Analyses (PRISMA) guidelines [44].

Studies were included in the review when they recruited people aged 50 years and above who were free living in the community, with either prevalence or incidence data on undernutrition, frailty, or sarcopenia. The data could be compared to those who were well-nourished, robust, or without sarcopenia. Studies included were observational studies, including cohort, cross-sectional, and case-controlled studies.

Studies were excluded that recruited participants from outpatient clinics, secondary care or those in hospitals, nursing homes or institutions. Studies where participants were recruited with any specific disease state (i.e., heart disease, diabetes, or stroke) or were from a specific group within society (i.e., army veterans, health workers, or dock workers) were excluded. In addition, studies were excluded which recruited people aged less than 50 years old. Excluded studies were systematic reviews, intervention studies, and qualitative works.

### 2.1. Search Strategy

A structured search was conducted of six electronic databases: Allied and Complementary Medicine (AMED), Cochrane Central Register of Controlled Trials (CENTRAL), Excerpta Medica database (EMBASE) via OVID, Medical Literature Analysis and Retrieval System Online (MEDLINE) via OVID, Web of Science, and Cumulative Index of Nursing and Allied Health Literature (CINAHL). The searches were performed in October 2020 using Medical Subject Headings (MeSH), using an asterisk at the end of keywords as a wild card to pick up all the possible words associated with each key term.

A librarian assisted in creating the search strategy, and the searches were amended as needed for each database. The authors used synonyms related to each keyword and then combined all keywords using the Boolean operators ‘OR’ and ‘AND’. There was no restriction on publication dates; the search was only limited to studies published in English and human studies. The following search terms were used: (Geriatrics AND Aged) AND (Malnutrition OR Anorexia OR Frail Elderly OR Sarcopenia) AND (Prevalence) OR (Incidence) AND (Limit to humans); see Appendix A.

After the searches were undertaken, the studies were uploaded to Covidence [45] by a reviewer (N.A.). Covidence was used to manage the data by identifying duplicate studies to remove [45]. All the identified studies were screened by title and abstract according to eligibility criteria by (N.A., A.A., J.S., A.M., and A.S.). The full-text of all selected abstracts was obtained and reviewed by one researcher (N.A.), with 20% independently reviewed by the second reviewer (S.B., C.T., and M.G.). A third person was asked to review to resolve any conflicts (S.B.). The eligibility criteria were modified after data extraction. This was because free-living was not defined clearly, so it was modified to improve clarity to exclude participants who had been recruited in outpatient clinics, hospitals, and any other institution.

### 2.2. Data Extraction

Data extraction from eligible studies was performed. A reviewer extracted the following information: general information (author, country, and publication year) and study characteristics (study design, sample size, methods used to assess undernutrition, frailty, and sarcopenia, funding source, key findings, and settings). Anthropometrics and body composition (height, weight, body mass index, fat mass, fat-free mass, and techniques used) were recorded. Functional measurements included handgrip strength, ‘time up and go’, and the walking test. Demographic information (age, gender, education, employment, and ethnicity) were recorded, along with the prevalence and incidence of undernutrition, frailty, and sarcopenia. The data were recorded on an Excel spreadsheet and a second reviewer checked 20% of the studies.

### 2.3. Quality Appraisal

The Joanna Briggs Institute Critical Appraisal Tools (JBI) were used to assess study quality using tools relevant to each study type. One researcher (N.A.) undertook the quality assessment, and a second person assessed 20% (S.B., C.T., and M.G.). The JBI tools consists of ten closed-ended questions (yes, no, unclear, and not applicable). These questions were designed to assess the study design, methodology, and analysis. The quality was categorized according to the total questions answered by yes as follows: high (total yes 10–8), medium (total yes 6–8), and low (total yes of 4–0). All studies with low quality were excluded from the review; for more information, see Appendix A.

### 2.4. Risk of Bias Assessment

Risk of bias was assessed using Risk of Bias In Non-Randomised Studies for Interventions (ROBINS-I), recommended by the Cochrane Handbook [46]. One author assessed the risk of bias with a second independent check of 20%.

### 2.5. Data Synthesis

Meta-analysis was conducted on relevant data for the prevalence of undernutrition, frailty, and sarcopenia, and subgroup analyses were undertaken on the tools used to measure the conditions and gender. For binary data, the prevalence and the total number in the sample used were extracted, and a DerSimonian random-effects model was used to calculate the effect size. The metaprop command within Stata was used to perform meta-analyses under binomial data. It computed 95% confidence intervals using the score statistic, and the exact binomial method incorporated the Freeman–Tukey double arcsine transformation for proportions. From this, weighted pooled estimates were produced.

Data were graphically displayed in Forest plots. The prevalence of undernutrition, frailty, and sarcopenia were presented as effect sizes with 95% confidence intervals (CI), in addition to the weights given to each study based on sample size.

An estimate of heterogeneity was performed using the Mantel–Haenszel model. I^2^ statistic values were calculated to quantify the degree of heterogeneity among studies, statistical heterogeneity was considered low if the I^2^ was <30% and heterogeneity was considered high if values were >50%.

## 3. Results

### 3.1. Search Results

The electronic database search identified a total of 1411 studies. After removing duplicates (62 articles), 1349 titles and abstracts were screened and 1195 studies excluded, leaving 154 full texts for a full screen against the eligibility criteria. The reasons for excluding studies at the full-text stage are shown in the PRISMA diagram (Figure 1). A total of 45 studies met the inclusion criteria and was assessed for quality. Of these, eight were excluded from the review due to low quality.

### 3.2. Study Characteristics

A total of 37 studies was included in the review, of which 20 were cross-sectional studies [15,47,48,49,50,51,52,53,54,55,56,57,58,59,60,61,62,63,64,65] and 17 were cohort studies [66,67,68,69,70,71,72,73,74,75,76,77,78,79,80,81,82]. Seven studies were undertaken in the United States of America [48,53,54,68,70,76,79], five each in Brazil [49,55,58,64,74], Japan [50,52,60,72,75], and China [56,62,73,77,81], three studies each in Mexico [47,51,59] and Korea [61,66,67], two studies in Spain [57,71], and one study in each of the following countries: Taiwan [82], Egypt [63], Italy [65], Germany [80], Australia [78], mixed European cities [15], and Belgium [69]; see (Appendix A Study Characteristics) for more details.

Seven studies measured a combination of undernutrition, frailty, or sarcopenia in older participants living in the community and reported the three conditions separately. Two studies in each case measured undernutrition and frailty [71,78], undernutrition and sarcopenia [56,62], and undernutrition, frailty, and sarcopenia together [59,61]. One study measured frailty with sarcopenia [50].

Two studies included undernourished people [63,65], four studies included older people with sarcopenia only [54,69,81,82], and twenty-four studies included older people who were frail only [15,47,48,49,51,52,53,55,57,58,60,64,66,68,70,72,73,74,75,76,77,79,80,83] (Table 1).

### 3.3. Participants

In all 37 included studies, there was a total of 106,041 participants, and the mean age reported was between 63.3 and 86.4 years old. Amongst the participants, there were 39,296 males and 48,518 females recruited. One study had 18,227 participants and did not provide details on gender.

### 3.4. Outcomes

#### 3.4.1. Prevalence

**Undernutrition** Five studies reported undernutrition in 4214 participants with an overall prevalence of 17% (95% CI 0.01, 0.46), with studies all given similar weighting of approximately 20%. The heterogeneity was high between the studies at I^2^ 99.7% (Figure 2). The prevalence of undernutrition in older people using the Mini Nutritional Assessment Short Form (MNA-SF) tool was higher, at 25% (95% CI 0.00, 0.70) reported by three studies [61,63,71], compared to a prevalence of 14% (95% CI 0.12, 0.16) reported by two studies which used the MNA tool [56,65] (Table 2 and Appendix A Prevalence of undernutrition analysis split by tools in the older people living in the community).

**Frailty** Twenty-eight studies measured the prevalence of frailty in 95,036 participants. The twenty-eight studies showed an overall prevalence of 13% (95% CI 0.11, 0.17) with similar weights within 3.38% to 3.64%. The heterogeneity was high between the studies at I^2^ 99.43% (Figure 3). The phenotype model was the most commonly used tool to identify frailty in 22 studies, showing a prevalence of 11% (95% CI 0.09, 0.13) [15,47,48,49,50,52,53,55,57,58,61,64,66,67,70,71,72,73,75,76,78,79]. However, one of these studies combined the phenotype model with Rockwood’s cumulative deficits model to identify frailty [58]. Other studies used the Rockwood’s cumulative deficit model [77], Edmonton Frail Scale validated for Brazilian Portuguese [74], 46 items in the frailty index [68], the frailty index with 31 deficits [51], the 25-item Kihon checklist [60], and the electronic frailty index [80] (Table 3 and Appendix A Prevalence of frailty analysis split by tools in the older people living in the community).

**Sarcopenia** Nine studies reported sarcopenia prevalence with 7656 participants, showing an overall prevalence of 14% (95% CI 0.09, 0.20). All studies had similar weighting, around 12%. The heterogeneity between the studies was high at I^2^ 97% (Figure 4). The nine studies used different tools. The highest rate was reported by four studies that used the Asian Working Group for Sarcopenia criteria (AWGS), with a prevalence of 16% (95% CI 0.07, 0.29) [50,56,61,62], whilst a rate of 12% (95% CI 0.08, 0.18) was reported by three studies that used the European Working Group on Sarcopenia in Older People criteria (EWGSOP) [54,69,81]. Two studies reported the lowest rate using the Simple Questionnaire to Rapidly Diagnose Sarcopenia (SARC-F), whose prevalence was 11% (95% CI 0.09, 0.13) [59,82] (Table 2 and Appendix A Prevalence of sarcopenia analysis split by tools in the older people living in the community).

**Combination of two or more conditions** Seven studies reported the prevalence of undernutrition, frailty, and sarcopenia in older participants living in community settings, which were presented separately and not included in the combined prevalence estimate of each condition (Table 3).

**Undernutrition and Frailty** Two studies measured both undernutrition and frailty [71,78]. However, both studies reported the prevalence of undernutrition and frailty separately. Lorenzo-lopes and colleagues showed 4% of participants were frail, and the number decreased during the follow-up to 3%. In contrast, 60 participants were malnourished at the baseline, and two were malnourished at follow-up time points [71]. In the second study, the nutritional status was reported as the BMI, with 5% of participants being underweight and 8% being frail [78] (Table 3).

**Undernutrition and Sarcopenia** Two studies measured both undernutrition and sarcopenia [56,62]. However, each study used the same tool to assess sarcopenia, namely, AWGS, but a different tool to assess undernutrition; Gao and colleagues used MNA-SF to measure undernutrition and Xu et al. used MNA [56,62]. Both studies compared malnourished participants in relation to sarcopenia. Goa et al. reported that the risk of developing sarcopenia would increase more than threefold when a participant was malnourished or at risk of undernutrition, using odds ratios (OR 3.53, 95% CI 1.68, 7.41) [62]. Whereas Xu et reported that 20% of participants were malnourished, with similar sarcopenia rates, 50.5% of participants had sarcopenia, compared to 49.5% without sarcopenia [56] (Table 3).

**Frailty and Sarcopenia** One study measured frailty and sarcopenia using the phenotype model and AWGS criteria [50]. The study reported that 4% of participants had both frailty and sarcopenia, compared to 2% who had frailty and 6% of participants who were reported to have sarcopenia [50] (Table 3).

**Undernutrition, Frailty, and Sarcopenia** Two studies measured undernutrition, frailty, and sarcopenia together [59,61]. Both studies used the phenotype model to identify frailty, while undernutrition and sarcopenia were assessed with different tools. One study used the MNA tool (74) and one used MNA-SF [61]. For sarcopenia, one study used the SARC-F scale [59] and one used AWGS [61]. The relationship between undernutrition, frailty, and sarcopenia was investigated by only one study identified [61]. This study reported the prevalence of frailty and sarcopenia combined was 60%, and frailty combined with the risk of undernutrition was found to have a prevalence of 83% [61] (Table 3).

#### 3.4.2. Incidence

Four studies reported incidence: three reported frailty and one sarcopenia in older people living in the community.

**Frailty incidence** Two studies used the phenotype model to assess frailty, while one study used the frailty index derived from the Rockwood cumulative deficits model. This study found the highest incidence of frailty with a standardised incidence, of 10% over one year in healthy people aged 55 years and above, and found the incidence increased with age and in females [77]. The second study also used the phenotype model and found that the frailty incidence was 6% in older men aged 75 years [78]. Comparing the results from both studies, the first study had younger participants of both genders and followed up over one year. In contrast, the latter study included men only, older participants aged 75 years and above, and followed up for three years. The third study used the frailty index and reported that the frailty status changed from robust to prefrail in 43% of the sample, and 8% of the sample changed from prefrail to frail [71]. Surprisingly, some participants showed some improvement in the frailty status: 33% of participants frail at baseline became prefrail during the follow-up period, and 9% of the prefrail participants became robust [71].

**Sarcopenia incidence** One study reported that the incidence of sarcopenia during a two-year follow-up was 6%, and the incidence after a four-year follow-up was 14% [81].

#### 3.4.3. Subgroup Analysis

There was a noticeable inequality in gender, which might lead to bias in results. Twenty-one studies had the ratio of male to female participants between 0.2 and 0.8, where there were more females than males [47,49,50,52,55,56,58,59,61,62,64,67,68,69,70,71,72,74,76,77,79]. However, one study [80] had a ratio of 1.32, which showed there were more male participants than females. Furthermore, one study included only the male gender [78], and another study did not provide information about the percentage of males and females [15]. The thirteen studies had similar numbers of participants in each gender, with a ratio ranging between 0.9 and 1 (Appendix A studies included in SR presented with female to male ratio). To overcome this issue, we conducted a subanalysis by gender.

**Undernutrition** The prevalence of undernutrition in females was 30% and 22% in males. When the prevalence was weighted for gender, the overall prevalence was 26% (95% CI 0.11, 0.44) (Appendix A prevalence of undernutrition analysis split by gender in the older people living in the community).

**Frailty** The prevalence of frailty in females was 17% and 12% in males, whilst the weighted prevalence for gender showed an overall prevalence of 15% (95% CI 0.12, 0.17) (see Appendix A prevalence of frailty analysis split by gender in the older people living in the community). These results included sixteen studies in females and seventeen in males. One study included males only, which explained the disparity in the number of participants [78]. The remaining eight studies did not report frailty according to gender [15,48,49,57,74,76,77,79].

**Sarcopenia** The prevalence of sarcopenia in females was 19% and in males was 11%; when the prevalence was weighted for gender, the overall prevalence was 15% (95% CI 0.09, 0.22) (Appendix A prevalence of sarcopenia split by gender in the older people living in the community).

### 3.5. Quality Appraisal

Forty-six studies were appraised using the JBI tools for cohort and cross sectional studies, and eight studies were excluded due to low quality. There were twenty-one cohort studies, seven of which were classified as strong [66,67,68,69,70,71,72] and ten as medium [73,74,75,76,77,78,79,80,81,82], while four studies were assessed as low-quality and were excluded [8,84,85,86]. There were twenty-four cross-sectional studies assessed, twelve of which were classified as strong [15,47,48,49,50,51,52,53,54,55,56,57], eight as medium [58,59,60,61,62,63,64,65], and four were found to be low-quality [87,88,89,90] (see Appendix A Joanna Briggs Quality Assessment).

### 3.6. Risk of Bias Assessment

The risk of bias was assessed using the Risk of Bias In Non-Randomised Studies for Interventions (ROBINS-I), recommended by the Cochrane Handbook [46]. All studies were graded critical to moderate, except one study graded as unclear (see Figure 5). Most of the studies included were identified as either having serious or critical levels of bias. The majority of this was due to missing data. This needs to be taken into account in the interpretation of the overall review as the level of certainty decreased. Further, high-quality prevalence and incidence studies need to be conducted.

## 4. Discussion

This systematic review and meta-analysis aimed to better understand the prevalence of undernutrition, frailty, and sarcopenia in older people living in the community. The overall prevalence of undernutrition was estimated to be 17% (95% CI 1% to 46%), which was within the range reported in other systematic reviews where prevalence ranged from 3.1% to 16.2% in older people living in the community [91,92,93]. The lowest prevalence of malnutrition in community settings was 3.1% (95% CI 2.3% to 3.8%) reported by a study that aimed to develop a quantitative summary of malnutrition prevalence measured by the MNA tool in different settings [88]. Similarly, a systematic review conducted on older people living in the community had a median malnutrition prevalence of 5% [92]. However, the purpose of the systematic review was to summarise the validity of different screening tools used to measure malnutrition prevalence [92]. The prevalence ranged from 0.5% to 22% when the MNA-Long form (MNA-LF) tool was used, and the MNA-SF tool had similar results to MNA-LF, which was 1.1% to 19% [92]. Furthermore, another study reported the prevalence of malnutrition as 16.2% in community-dwelling older people aged ≥65 years that measured the nutritional status using the Short Nutritional Assessment Questionnaire 65+ (SNAQ65+), [92]. Both the MNA and MNA-SF were effective tools for screening in older people who were at risk of malnutrition in hospitals [12] and primarily included questions on changes over a short period of time, meaning more longer time changes in community settings may not have been identified.

The prevalence of frailty in this review was 13%, similar to the available literature. One previous review cited a slightly lower prevalence, with older people living in the community having an overall prevalence of 10.7% across 21 studies [94]. It aimed to compare and summarise the frailty prevalence reported by age, gender, and tool used. Thus, the rate increased to 13.6% once weighted for the phenotype model [94], which was comparable to the findings of the current review as majority of the included studies used the phenotype model.

For sarcopenia, the prevalence from the present review was 14%, which was slightly higher than previous data reported. A systematic review found the AWGS tool used to measure sarcopenia prevalence from eight studies of men and nine studies of women showed a prevalence rate of 9.8% and 10.1%, respectively [95]. Similarly, a recent review including 41 studies concluded that the prevalence of sarcopenia was 11% in males and 9% in females among people from Europe (UK, Italy, Spain, Belgium, Germany, Denmark, and Turkey), Asia (China, Japan, Taiwan, Iran, and Korea), South American (Chile), and Africa (Gambia) living in the community [96]. These studies used different tools to identify sarcopenia, including EWGSOP, AWGS, dual-energy x-ray absorptiometry (DEXA), the bioelectrical impedance analysis (BIA), equation of total muscle mass (TMM), anthropometric measurements, and anthropometric equation [97]. In contrast, a systematic review that included six studies using the EWGSOP tool showed a wide-ranging prevalence from 8.8% to 36.5% among older people living in the community [98].

These differences in prevalence may be related to the different tools used to identify undernutrition, frailty and sarcopenia, which concurred with our findings. For undernutrition, the MNA-SF tool detected more cases than the MNA tool. A review that compared MNA to MNA-SF showed a higher sensitivity and specificity to the full MNA [99].

The prevalence of sarcopenia was found to be higher using AWGS compared to the SARC-F and EWGSOP tools. In contrast, one review compared eight methods of identifying sarcopenia, with the lowest prevalence rates from AWGS/EWGSOP (12.9%), the International working Group on Sarcopenia (9.9%), and the Foundation for the National Institutes of Health definitions (18.6%). Other methods showed a higher prevalence, including: appendicular lean mass (ALM)/weight (40.4%), ALM/height (30.4%), ALM regressed on height and weight (30.4%), and ALM/BMI (24.2%) definitions [100].

Another reason for the variation between prevalence rates may have been due to gender-specific differences. The present systematic review found that prevalence rates changed after a subgroup analysis for each gender, with undernutrition being higher in females than males, comparing the gender subgroup prevalence of 26% with the overall prevalence for the population of 17%. Overall, the findings showed that undernutrition prevalence increased nearly two-fold after considering gender, which is reasonable given the variation in body composition between males and females [101,102]. Similarly, our results revealed that frailty occurred slightly more frequently in females than males. The subgroup prevalence for gender of 15% was somewhat higher than the finding for the overall prevalence of the population, in which the result was 13%. Thus, when considering gender, the overall prevalence increased. These results showed that undernutrition and frailty were more common in females than males. However, in this review, there was no significant difference between the overall prevalence of sarcopenia in the total population (14%) compared to when considering gender (15%). Our findings were consistent with the literature, showing that the prevalence of undernutrition [103] and frailty [94,104] varied according to gender, with a markedly increased rate in females compared to males, except for sarcopenia [96]. This systematic review looked at age and gender. However, there are other factors that could actually affect the older population, which include psychosocial (such as mental health, loneliness, and social isolation), environmental, economic (including access to food, food preparation, and neighbourhood walkability), and physiologic factors (for example oral health and medication use) [105]. When comparing the prevalence of undernutrition, frailty, and sarcopenia in hospital settings, 66% was obtained for undernutrition, 47% for frailty, and 37% for sarcopenia in older adults, which was considerably higher than data reported in this present review for community-dwelling adults [106].

The second key finding was a wide range of results for the prevalence of two conditions or more in older people living in the community. The majority of included studies found the prevalence was greater for participants with one or more of undernutrition, frailty, or sarcopenia combined, compared to the prevalence indicated for robust participants or those of a healthy weight or not sarcopenic. One study found older people with sarcopenia had a lower BMI of 21.6 kg/m^2^ (SD 3) and had lower MNA scores of 11.6 (SD 1.9), which indicated that people were at risk of malnutrition compared to older people without sarcopenia with a BMI of 23.1 kg/m^2^ (SD 3.2) and MNA score of 12.4 (SD 1.7) [62]. One study looked at frailty along with the burden of other geriatric conditions and found that frail participants were more at risk of undernutrition (83.3%), followed by sarcopenia (60.3%) compared to robust participants with sarcopenia (21.8%), and at risk of malnutrition (45.5%) [61]. The authors also found more frail participants with sarcopenia and at risk of undernutrition compared to robust participants with sarcopenia and at risk of undernutrition [61]. One study reported sarcopenia prevalence with older participants at risk of undernutrition or malnourished higher around three times than those well-nourished and with sarcopenia (50.5% vs. 18.7%) [56]. However, one study observed that fewer older participants had frailty and sarcopenia coexisting than robust participants without sarcopenia [50]. These differences might be related to gender, age, and the tool used to define the conditions. These three factors may have influenced the prevalence estimates, as shown in our systematic review for undernutrition, frailty, and sarcopenia. A few studies that looked at two conditions together showed little agreement between the studies [50,56,59,61,62,71,78]. Thus, we could not draw any conclusions, indicating further research is required to understand the interdependency between the nutritional status and frailty [36].

Few studies have measured the incidence of these conditions. Of those that have, they also showed the rate increased with age. The low number of studies measuring incidence might be due to the study design, requiring a long follow-up which is costly and has a high rate of participants lost to follow up [107]. This systematic review included three studies that measured the frailty incidence and reported an increased number of frailty cases with an increasing age over the follow-up period [71,77,78]. Similarly, the study that looked at the incidence of sarcopenia showed an increasing rate of cases over a range of follow-up periods from two to four years [81]. Our findings were consistent with previous studies that showed undernutrition, frailty, and sarcopenia increased with ageing [37,108,109,110].

Interestingly, one study included in the review found participants’ frailty status changed in reverse, from frail to prefrail, and those prefrail participants became robust [71]. This result indicates that frailty status can improve with treatment interventions. These approaches could be a combination of physical exercises and nutritional interventions [111].

This systematic review had a few limitations. Only one researcher carried out the search. However, an expert librarian helped with the search strategy. Another limitation was that the second independent reviewer processed only 20% of the studies when assessing their quality and reviewing the data extraction. These identified studies had different measurement tools that restricted the pooling of data and meta-analysis. It is also noteworthy that this review included the association between physical frailty and undernutrition, although there were no studies included that identified psychosocial frailty, which is a further limitation. The heterogeneity across studies, particularly for undernutrition, was high with large ranges observed between studies for prevalence. These studies used different tools to measure undernutrition in different countries and in rural or urban environments from samples with different socioeconomic statuses, which may explain some of the variation.

The strengths of this review were that multiple databases were searched for studies, and meta-analyses were possible for prevalence data, including undernutrition, frailty, and sarcopenia, with a large number of participants included from the studies.

## 5. Conclusions

All studies included in the systematic review measured the prevalence of undernutrition, frailty, or sarcopenia, yet very few estimated the incidence. No study measured the prevalence of the three conditions in the same participants or reported the overlap between the conditions. The current findings showed a lack of evidence measuring the overlap between undernutrition, frailty, and sarcopenia in older people. Finding the relationship between the three conditions and their overlaps would require measuring the incidence or prevalence within the same cohort of older people. Further research is needed to evaluate the relationship between undernutrition, frailty, and sarcopenia in older people living in community dwellings and to determine the effectiveness of diet and activity interventions for treating or preventing any of these conditions in community-dwelling people over the age of 50.

## Figures and Tables

**Figure 1 nutrients-14-01537-f001:**
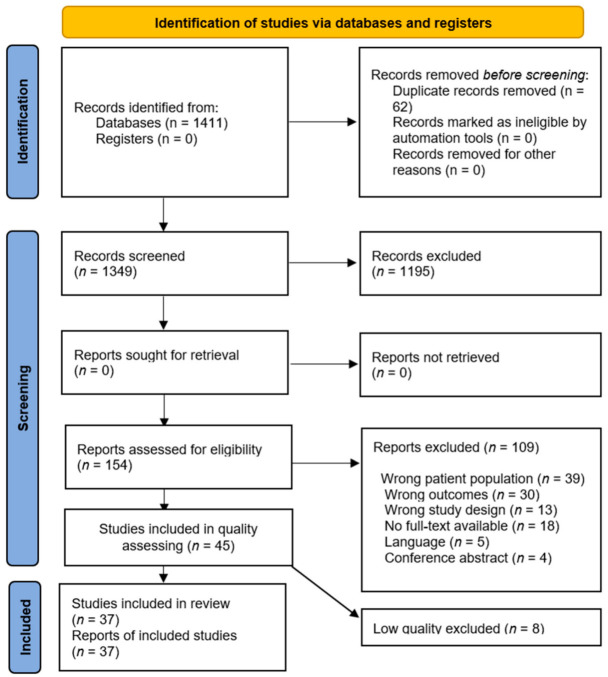
PRISMA flow diagram used to record the selection process in the current research.

**Figure 2 nutrients-14-01537-f002:**
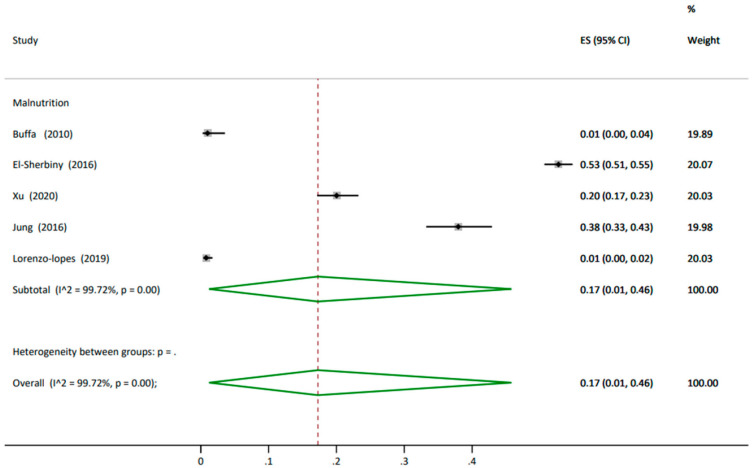
Prevalence of undernutrition in the older people living in the community. ES: effect size; CI: confidence interval.

**Figure 3 nutrients-14-01537-f003:**
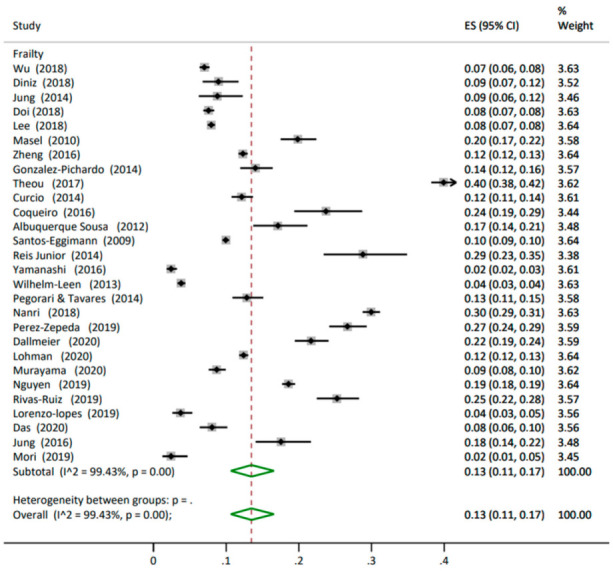
Forest plot for the prevalence of frailty in older people living in the community. ES: effect size; CI: confidence interval.

**Figure 4 nutrients-14-01537-f004:**
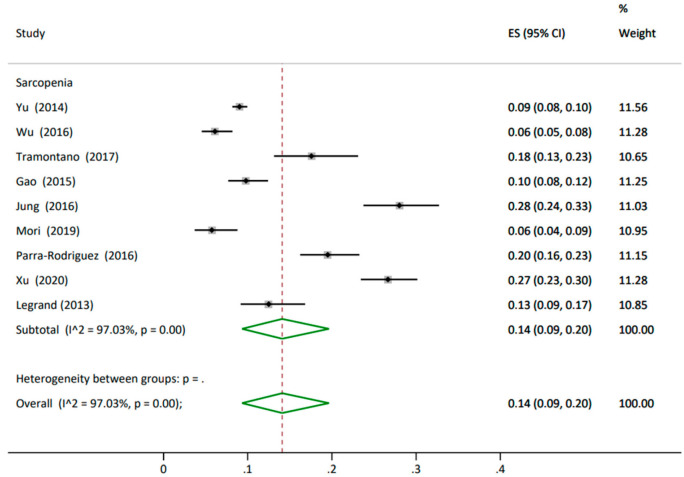
Prevalence of sarcopenia in older people living in the community. ES: effect size; CI: confidence interval.

**Figure 5 nutrients-14-01537-f005:**
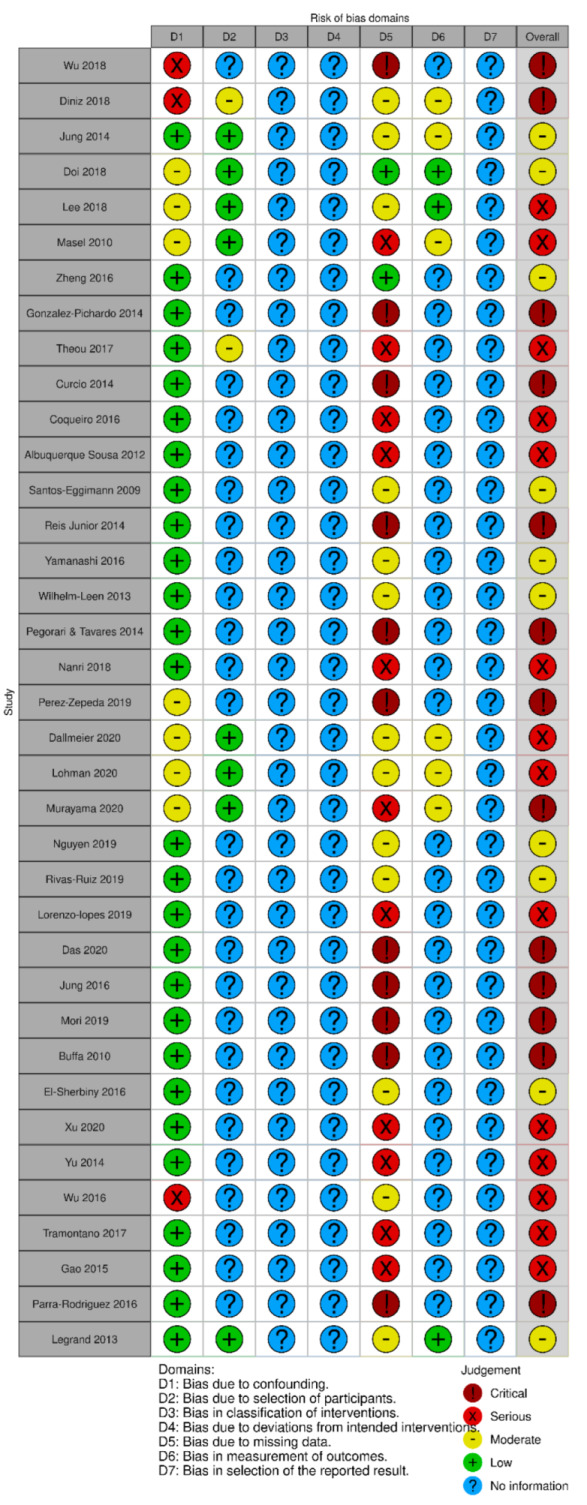
Risk of bias summary for each included study.

**Table 1 nutrients-14-01537-t001:** Study characteristics.

Author (Year)	Sample *N*	Age in Years Mean (SD)	Gender *n* (%)	BMI kg/m^2^ Mean (SD)
Undernutrition				
Buffa (2010) [65]	200	81 (7)	M 100 (50) F 100 (50)	26.6 (3)
El-Sherbiny (2016) [63]	2219	69 (7)	M 1165 (53) F 1054 (48)	
Frailty				
Wu (2018) [73]	5301	60≤	M 2682 (51) F 2619 (49)	
Diniz (2018) [74]	T1 515 T2 262	T1 75 (7) T2 79 (6) Death 79 (8)	T1:M 168 (33) F 347 (67) T2:M 88 (34) F 174 (66)	
Doi (2018) [75]	4676	R 71 (4) PF 72 (6) Frail 77 (6)	M 2316 (50) F 2360 (51)	R 24 (3), PF 24 (3), Frail 23 (4)
Jung (2014) [66]	341	LMI, Median (IQR): No Decline 69 (72–67) Decline 70 (74–67)	M 178 (52) F 163 (48)	LMI, Median (IQR): No Decline 24 (22–26) Decline 25 (23–27)
Lee (2018) [67]	11,266	73 (7)	M 4540 (40) F 6726 (60)	
Masel (2010) [76]	1008	82 (5)	M 371 (37) F 637 (63)	N (%): underwt 13 (1), healthy 332 (33), overwt 404 (40), obese 259 (26).
Zheng (2016) [77]	10,039	71 (8)	M 3885 (39) F 6154 (61)	
Gonzales-Pichardo (2013) [47]	927	78 (6)	M 418 (45) F 509 (55)	
Theou (2017) [68]	3141	63 (10)	M 1426 (45) F 1715 (55)	
Curcio (2014) [48]	1878	71 (7)	M 897 (48) F 981 (52)	24 (5)
Coqueiro (2017) [49]	316	74 (10)	M 143 (45) F 173 (55)	-
Albuquerque (2012) [64]	391	74 (7)	M 151 (37) F 240 (61)	-
Santos-Eggimann (2009) [15]	18,227	Two grps 50–64 and ≥65	-	-
Reis Junior (2014) [55]	236	Three grps 60–69, 70–79 ≥80	M 97 (41) F 139 (59)	N (%): Frail-Underwt 22 (29), Healthywt34 (27), Overwt 10 (13) PF-Underwt 46 (61), Healthywt 64 (50) Overwt 57 (71)
Yamanashi (2016) [52]	1818	72.2 (8)	M 662 (36) F 1156 (64)	23 (3)
Wilhelm-Leen (2013) [53]	4667	60≤	M 2288 (49) F 2379 (51)	-
Pegorari (2014) [58]	958	74 (7)	M 341 (36) F 617 (64)	-
Nanri (2018) [60]	5638	M 73 (6) F 74 (6)	M 2707 (48) F 2931 (52)	M 23 (3) F 23 (3)
Perez-Zepeda (2019) [51]	1128	69 (8)	M 550 (49) F 578 (51)	28 (5)
Dallmeier (2020) [80]	1204	Median (IQR1,3) M 74 (70,82) F 73 (70, 80)	M 692 (58) F 512 (43)	M 28 (4) F 27 (5)
Lohman (2020) [70]	10,490	R 71 (0.1), PF 74 (0.2) Frail 79 (0.3)	M 4579 (44) F 5914 (56)	-
Murayama (2020) [72]	2206	Five grps: (65–69), (70–74), (75–79), (80–84), (85≤)	M 962 (44) F 1244 (56)	-
Nguyen (2019) [79]	7197	Six grps: 65–69 392 (19), 70–74 1541 (21), 75–79 1461 (20), 80–84 1422 (20), 85–89 859 (12), >90 522 (7)	M 3050 (42) F 4147 (58)	-
Rivas-Ruiz (2019) [57]	855	78 (5)	M 402 (47) F 453 (53)	29 (4)
Sarcopenia				
Yu (2014) [81]	4000	73 (5)	M 2000 (50) F 2000 (50)	24 (3)
Wu (2016) [82]	670	76 (6)	M 340 (51) F 330 (49)	24 (3)
Tramontan (2017) [54]	222	≥65	M 102 (46) F 120 (54)	Sarcopenia 21.1 (2.3) w/o Sarcopenia 24 (4.2)
Legrand (2013) [69]	288	84.8 (3.6)	M 103 (35.8) F 185 (64.2)	Sarcopenia 23 (4.1) w/o Sarcopenia 27.8 (4.2)
Combination of two or more of undernutrition, frailty or sarcopenia				
Xu (2020) [56]	664	86 (4)	M 281 (42) F 383 (58)	n(%)-Underwt 33 (5) Healthywt 299 (45) Overwt 256 (39) Obese 76 (12)
Gao (2015) [62]	612	Healthy 71 (6) Sarcopenia 77 (7)	M 254 (42) F 358 (58)	Sarcopenia 22 (3) w/o Sarcopenia 23 (3)
Jung (2016) [61]	382	74 (7)	M 167 (44) F 215 (56)	-
Mori (2019) [50]	331	72 (5)	M 93 (28) F 238 (72)	23 (3)
Parra-Rodriguez (2016) [59]	487	73 (8)	M 97 (20) F 390 (80)	28 (5)
Lorenzo-lopes (2019) [71]	T1 749 T2 537	T1 76 (7) T2 76 (7)	T1 M 295 (39) F 454 (61) T2 M 206 (38) F 331 (62)	MNA-SF n (%): T1 healthy 642 (86), at-risk 101 (14), maln 6 (0.4) T2 healthy 472 (88), at-risk 62 (12), maln 2 (0.4)
Das (2020) [78]	T1 794 T2 781	81 (5)	M 794 (100) F 781 (100)	28 (4)

BMI—body mass index; F—female; healthywt—healthy weight; IQR—interquartile range; LMI—lean mass index; MNA-SF: Mini Nutritional Assessment Short Form (ranges 0–14 points; 12–14 points indicate normal nutritional status; 8–11 points indicate risk of undernutrition; 0–7 points indicate undernutrition); M—male; Overwt—overweight; PF—prefrail; R—robust; T1—baseline; T2—follow-up; Underwt—underweight.

**Table 2 nutrients-14-01537-t002:** Prevalence in studies of older people with undernutrition, frailty, or sarcopenia.

Author (Year)	Prevalence			
Undernutrition	Tools Used	Normal *n* (%)	At Risk *n* (%)	Undernutrition *n* (%)
Buffa (2010) [65]	MNA	107 (63)	61 (36)	2 (1)
El-Sherbiny (2016) [63]	MNASF	-	-	<70 years 648 (48) >70 years 522 (60)
Frailty	Tools used	Robust *n* (%)	Prefrailty *n* (%)	Frailty *n* (%)
Wu (2018) [73]	Physical Frailty Phenotype	2216 (42)	2714 (51)	371 (7)
Diniz (2018) [74]	Edmonton Frail Scale	T1 156 (59) T2 75 (29)	T1 60 (23) T2 55 (21)	T146 (18) T2 58 (46)
Doi (2018) [75]	Fried Phenotype	1978(42)	2344 (50)	354 (8)
Jung (2014) [66]	Fried Phenotype and LMI	No LMI 67 (43) LMI 59 (32)	No LMI 81 (52) LMI 104 (57)	No LMI 9 (5) LMI 21 (11)
Lee (2018) [67]	Fried Phenotype	4654 (41)	5716 (51)	896 (8)
Masel (2010) [76]	Fried Phenotype	263 (26)	545 (54)	200 (20)
Zheng (2016) [77]	Cumulative Deficit	8803 (88)	-	1236 (12)
Gonzales-Pichardo (2013) [47]	Fried Phenotype	450 (49)	347 (37)	130 (14)
Theou (2017) [68]	46-item FI	766 (24)	1121 (36)	1254 (40)
Curcio (2014) [48]	Fried Phenotype	654 (35)	996 (53)	228 (12)
Coqueiro (2017) [49]	Fried Phenotype	241 (76)	-	75 (24)
Albuquerque (2012) [64]	Fried Phenotype	89 (23)	235 (60)	67 (17)
Santos-Eggimann (2009) [15]	Fried Phenotype	50–64 years 5308 (59) 65+ years 3056 (41)	50–64 years 3394 (37) 65+ years 3177 (42)	50–64 years 372 (4) 65+ years 1277 (17)
Reis Junior (2014) [55]	Fried Phenotype	41 (18)	139 (59)	56 (24)
Yamanashi (2016) [52]	Fried Phenotype	1139 (63)	635 (35)	44 (2)
Wilhelm-Leen (2013) [53]	Fried Phenotype	M 2227 (97) F 2262 (95)	-	M 61 (3) F 117 (5)
Pegorari (2014) [58]	Fried Phenotype	M 122 (39) F 191 (61)	M 187 (36) F 353 (64)	M 32 (26) F 91 (74)
Nanri (2018) [60]	Kihon Checklist	M 1241 (46) F 1295 (44)	M 678 (25) F 735 (25)	M 788 (29) F 901 (31)
Perez-Zepeda (2019) [51]	31-item FI	827 (73)	-	301 (27)
Dallmeier (2020) [80]	eFI	-	-	M 128 (18.5) F 133 (26)
Lohman (2020) [70]	Fried Phenotype	3934 (38): M 1795 (50) F 1774 (50)	5255 (50): M 2295 (43) F 3093 (57)	1301 (12): M 489 (32) F 1044 (68)
Murayama (2020) [72]	Fried Phenotype	1279 (50)	1033 (41)	220 (9)
Nguyen (2019) [79]	Fried Phenotype	2213 (31)	3647 (51)	1337 (19)
Rivas-Ruiz (2019) [57]	Fried Phenotype	-	-	216 (26)
Sarcopenia	Tools used	w/o Sarcopenia *n* (%)		Sarcopenia *n* (%)
Yu (2014) [81]	EWGSOP Criteria	-		361 (9)
Wu (2016) [82]	SARC-F	629 (94)		41 (6)
Tramontan (2017) [54]	EWGSOP Criteria	183 (82)		39 (18)
Legrand (2013) [69]	EWGSOP Criteria	252 (88)		36 (12)

eFI—electronic frailty index; EWGSOP—The European Working Group on Sarcopenia in Older People; F—female; FI—frailty index; LMI—lean mass index; M—male; MNA—mini nutritional assessment; MNA-short—mini nutritional assessment short; MUST—malnutrition universal screening tool; MST—malnutrition screening tool; MNA-SF—mini nutritional assessment short; MNA-LF—mini nutritional assessment long; SARC-F—Simple Questionnaire to Rapidly Diagnose Sarcopenia; T1—baseline; T2—follow-up; w/o = without; 2 yrf/u—two-year follow-up; 4 yrf/u—4-year follow up.

**Table 3 nutrients-14-01537-t003:** Prevalence and incidence in studies of people with the combination of two or more undernutrition, frailty, or sarcopenia.

Authors (Year)	Tool Used	Nutrition Status *n* (%)	Frailty *n* (%)	Sarcopenia *n* (%)	Combination *n* (%)
Xu (2020) [56]	MNA AWGS	Normal 466 (80) Under 116 (20)	-	S 427 (73) S 155 (27)	Normal + S 29 (19) Normal no S 347 (81) Under + S 78 (51) Under no S 211 (50)
Gao (2015) [62]	MNA-SF AWGS criteria	-	-	No S 552 (90) S 60 (10)	MNA no S M (SD)12.4 (2) MNA + S M (SD) 11.6 (2)
Jung (2016) [61]	MNA-SF Fried Phenotype AWGS	Under 145 (38)	R 76 (20) Prefrailty 201 (53) Frailty 67 (17)	S 107 (28)	No frailty and S 69 (22) No frailty and at risk 143 (46) Frailty and S 40 (60) Frailty and at risk 55 (83)
Mori (2019) [50]	Fried Phenotype AWGS	-	R 292 (88) Frailty 8 (2)	S 19 (6)	S or frailty 39 (12) S and frailty 12 (4)
Parra-Rodriguez (2016) [59]	MNA Fried Phenotype SARC-F	MNA, M (SD) 25 (3)	M (SD) 1 (1)	M(SD) 2 (2): No S 392 (81) S 95 (20)	-
Lorenzo-lopes (2019) [71]	MNA-SF Fried Phenotype	T1: Healthy 642 (86) At-risk 101 (14) Maln 6 (0.4)	T1: R183 (24) Prefrailty 538 (72) Frailty 28 (4)	-	-
Das (2020) [78]	BMI Fried Phenotype	Under 42 (5) Healthy 506 (65) Over 200 (26)	T1R 341 (44) Prefrailty 362 (47) Frailty 64 (8)	-	-

AWGS criteria—Asian working group for sarcopenia; BMI—body mass index; MNASF—Mini-Nutritional Assessment Short Form; MNA score—mini nutritional assessment; MNA-SF—mini nutritional assessment short form; Over—overweight; R—robust; SARC-F—Simple Questionnaire to Rapidly Diagnose Sarcopenia; S—sarcopenia; T1—baseline; Under—underweight; R—robust.

## Data Availability

The data that support the findings of this study are available on request from the corresponding author.

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
