# Peer review of "Prevalence of Undernutrition, Frailty and Sarcopenia in Community-Dwelling People Aged 50 Years and Above: Systematic Review and Meta-Analysis"

_nutrients, 2022, doi:10.3390/nu14081537_

Round 1
Reviewer 1 Report
Comments to the Author
This is a practically important review article indicates that Prevalence of Undernutrition, Frailty and Sarcopenia in Community-Dwelling Older People: Systematic Review and Meta-analysis.
However, it is necessary to reexamine the research method etc. in several respects.
- Title
In the previous study, the age of the participants was over 50 years old.
Modify from "older" people to "people aged 50 years and above".
- Table 1. Study Characteristics
A recent review including 41 studies concluded that the prevalence of sarcopenia was 11% in males and 9% in females among European, Chinese, Japanese, Taiwanese, British, Turkish and Korean people living in the community [97].
Add the following information.
Asia or Europe, country name, etc.
- Table 1. Study Characteristics
In the UK, approximately three million people are undernourished [7], and the likelihood of occurrence increases with age and comorbidities [8].
If possible, add the rates of comorbidities.
- Discussion
Gerdien C et al. presented the following paper in hospitalized older adults.
Frailty, Sarcopenia, and Malnutrition Frequently (Co-)occur in Hospitalized Older Adults: A Systematic Review and Meta-analysis
J Am Med Dir Assoc. 2020 Sep;21(9):1216-1228. doi: 10.1016/j.jamda.2020.03.006.
Compared to hospitalized older people, community-dwelling older people would have higher nutritional status and lower rates of sarcopenia and frailty.
Please describe the clinical characteristics of sarcopenia, frailty, and low nutrition of community-dwelling elderly in your discussion. (In particular, differences from hospitalized elderly.)
- Results, and Discussion
Please show the results and discuss association between sarcopenia, or Frailty and (Risk of) undernutrition.
For example, the odds ratio of sarcopenia (or Frailty) in the presence of (risk of) undernutrition relative to the absence of (risk of) malnutrition was xx (95% CI: xx, xx) in the community-dwelling people.
- Please show the discuss clinical relevance of findings and future research in discussion.
- Figure 5
Figure 5has low resolution.
- Discussion
Are the MNA-SF and MNA appropriate for assessing the nutritional status of community-dwelling people?
- Discussion
Currently, no accepted reference standard exists to identify frailty, and extensive international efforts are underway to identify the means of optimal measurement. Three major approaches to defining frailty exist:
(1) The physical phenotype model of Fried et al.
L.P. Fried, C.M. Tangen, J. Walston, et al.Frailty in older adults: Evidence for a phenotype
J Gerontol A Biol Sci Med Sci, 56 (2001), pp. M146-M156
(2) The deficit accumulation model of Rockwood and Mitnitski which captures multimorbidity
A.B. Mitnitski, A.J. Mogilner, K. Rockwood. Accumulation of deficits as a proxy measure of aging. Sci World J, 1 (2001), pp. 323-336
(3) Mixed physical and psychosocial models, such as the Tilburg Frailty Indicator and Edmonton Frailty Scale
R.J. Gobbens, M.A. van Assen, K.G. Luijkx, et al. The Tilburg Frailty Indicator: Psychometric properties. J Am Med Dir Assoc, 11 (2010), pp. 344-355
D.B. Rolfson, S.R. Majumdar, R.T. Tsuyuki, et al.Validity and reliability of the Edmonton Frail Scale. Age Ageing, 35 (2006), pp. 526-529
In this study, there are many references that were evaluated in the Fried Phenotype model. Therefore, you were able to report on the association between physical frailty and undernutrition. However, you were not able to clarify the relationship between psychosocial frailty and undernutrition.
Author Response
The authors would like to thank the reviewers for their insightful comments that have improved the quality of our manuscript.
Reviewer 2 Report
The article entitled “Prevalence of Undernutrition, Frailty and Sarcopenia in Community-Dwelling Older People: Systematic Review and Meta-analysis” examined the evidence on the prevalence and incidence of undernutrition, frailty and sarcopenia in older people aged >50 years living independently in community dwellings. I think the topic could be of interest to the readers of Nutrients. However, the article has some major problems as follows: 1. First of all, something is wrong with the references. Some of them are not presented at the main text. Thus, the evaluation of the whole study is difficult. 2. Abstract: I suggest to introduce the number of the PROSPERO registration. 3. Key words: Please add to the present key words: ‘systematic review’ and ‘meta-analysis’. 4. Introduction: a. In my opinion the introduction should be updated. For example, there is a relatively new systematic review and meta-analysis about these issues among hospitalized elders: Ligthart-Melis, Gerdien C., et al. "Frailty, sarcopenia, and malnutrition frequently (co-) occur in hospitalized older adults: a systematic review and meta-analysis." Journal of the American Medical Directors Association 21.9 (2020): 1216-1228. 5. Methods a. I think the ‘Search strategy, inclusion and exclusion criteria’ subsection is needed, where the authors should provide the details of exclusion and inclusion criteria. What are the main criteria in details? What does it mean that someone was excluded because it had a specific disease? What kind of articles did you include to the systematic review and meta-analysis (case reports, original articles, dissertations, books, etc)? I suggest to describe what kind of screening tools/scales should be in the included papers to assess frailty or sarcopenia or undernutrition. b. The authors indicate that they finished searching on October, 2020. Thus, I suggest to search once again the databases up to date. c. The authors should also provide the searching terms in the search strategy section in details. d. The authors showed that they prepare flow chart based on PRISMA chart. Nevertheless, there is a lot of mistakes in counting. For example, step ‘records after duplicates removed’ means how many records were assessed without duplicates. Secondly, there is a new PRISMA flow diagram (2020) which should be addressed in final version. e. The searching process in the first paragraph of ‘Study selection’ is not sufficient. Please provide who was the searcher, who resolved whole disagreements between those two researchers etc. What about the missing data, did you try to contact with the authors? f. Description of the quality assessment should be more advanced, and the references should be provided. What was the cut-point for the inclusion? 6. Results a. There is a lot of mistakes in counting at the presented ‘Search results’ and “prisma flow diagram’. For example, there is a lack of explanations for two records - why did you exclude them at the “full-text articles excluded, with reasons” step. b. I think the table 1 should be more advanced – the additional descriptions are needed and it should be like: study, country, study design, population, number of participants, gender, measure/indicator of frailty/undernutrition/sarcopenia, main findings, and quality score. 7. General comments a. Authors should complete or rewrite the issues as was suggested. Moreover, they should pay attention to the accurate demonstration of the results and try to organize the whole manuscript according to guidelines such as PRISMA or Cochrane. It seems that the authors do not follow point by point the instructions to obtain well-organized systematic review and meta-analysis. b. The crucial issue that there is a lot of mistakes at the results presentation, which should be interpreted with caution.Author Response
The authors would like to thank the reviewers for their insightful comments that have improved the quality of our manuscript.

Reviewer 3 Report
This paper set out to review of published literature on the incidence and prevalence of undernutrition, frailty and sarcopenia in community dwelling adults over the age of 50. It also aimed to look at these conditions by sex and to look at the overlap of these three conditions in the same populations. While it reported information about prevalence, there was little data on incidence and essentially no data reporting on the three conditions occurring in the same populations. Most of the findings reported were consistent with existing literature, although there was high heterogeneity and high potential bias associated with the total data set evaluated.
Because there were few new findings compared to existing literature and because there was insufficient information on incidence and prevalence of the three conditions of interest occurring in the same population, the introduction and discussion would benefit from additional explanation about what is novel about this review. Did this review cover studies that have not previously been reviewed? Are there important new additions to the literature that have occurred since the last published review and what are those important new findings? Does this paper review studies that used newer methods (and if so, what are they)? If the major finding of this review is that no studies have yet measured all three conditions of interest in the same individuals, then that should be more prominently drawn out in the discussion. Bottom line: What is the impact of this paper given that most of the findings as presented are similar to previous reports?
Some additional comments are listed here:
Line 101: Why did the authors choose to study ages 50 and above vs. a lower age cutoff that is higher and more consistent with other studies in the literature? How might the selection of this lower age limit have affected the results? How many studies were included because of this that would otherwise have been excluded had a higher age limit have been selected…e.g., above age 60?
Lines 104-107: Why were people visiting outpatient clinics not included? They could be perfectly healthy and merely visiting for an annual wellness check. And, what is an example of a “particular group within society” that were omitted?
Lines 166-168 and Figure 1: Some of the numbers do not add up. For example, 1411 records were identified and 62 were duplicates that were removed (the box with 62 duplicates removed should show 1349 records). The next box shows 1398 records screened…shouldn’t this be 1349, i.e., the total number minus the duplicates removed? Then, 1247 records were removed…this should leave either 1349-1247=102, OR if 1398 really is the correct number screened the box should show 1398-1247=151, not 154 as shown. The math in the text and on the consort diagram should be corrected and should reflect the actual numbers of records selected and reviewed.
Lines 198-200: The authors state that there were a total of 104,398 participants with 39,296 males and 48,518 females. 39,296 + 48,518 = 87,814, not 103,398. Which of these numbers is correct?
Line 311 and throughout: Is it possible for the authors to provide greater detail about the nature of the “undernutrition” reported in the various studies? What nutrients, sources etc. were most at risk in the populations studied? Likewise, for frailty, were there particular aspects of frailty that were most prevalent? Adding some specificity here may help distinguish this paper from the many other papers reporting similar prevalence results.
The discussion would benefit from additional insight about what readers should take away from this analysis. Given such high heterogeneity across studies it begs the question of why was heterogeneity so high? Is it more due to assessment methods used, different populations, different contextual conditions among the populations? What should the reader take away that is meaningful?
Likewise, the bias assessment revealed that 36 of the 46 studies were at risk of either serious or critical bias with the remaining 10 at risk of moderate bias, all mostly due to missing data. What does this say about the overall results and conclusions? The authors could help the reader understand what the take home message is given what appears to be high potential bias.
Finally, do the authors have anything to say about whether any of the conditions measured (undernutrition, frailty, sarcopenia) are influenced by contextual features surrounding the subjects in the studies reviewed, like the local living environment, resource availability, access to healthy food, neighborhood walkability, etc. In other words, how much of the prevalence for the conditions studied was due to age and sex vs. the various psychosocial and sociodemographic factors that associate with health? This may not be available from the studies reviewed but probably needs to be called out as missing information that could explain such large differences in condition prevalence across the studies reviewed and should be a focus in future studies aimed at understanding these conditions.
Author Response

(The authors gave the same response as above.)

Round 2
Reviewer 1 Report
No comment
Reviewer 2 Report
I accept th paper in the present form
Reviewer 3 Report
The authors have addressed the main concerns raised by the reviewers and the manuscript has been improved. There are still some minor English usage/spelling and punctuation errors that require editorial attention.